# Distinct Effects of Chemical Toxicity and Radioactivity on Metabolic Heat of Cultured Cells Revealed by “Isotope-Editing”

**DOI:** 10.3390/microorganisms11030584

**Published:** 2023-02-25

**Authors:** Jana Oertel, Susanne Sachs, Katrin Flemming, Muhammad Hassan Obeid, Karim Fahmy

**Affiliations:** 1Helmholtz-Zentrum Dresden-Rossendorf, Bautzner Landstrasse 400, 01328 Dresden, Germany; 2Protection and Safety Department, Atomic Energy Commission of Syria, Damascus P.O. Box 6091, Syria; 3Cluster of Excellence Physics of Life, Technische Universität Dresden, 01062 Dresden, Germany

**Keywords:** bacteria, growth rate, isothermal microcalorimetry, low dose, metabolic monitoring

## Abstract

Studying the toxicity of chemical compounds using isothermal microcalorimetry (IMC), which monitors the metabolic heat from living microorganisms, is a rapidly expanding field. The unprecedented sensitivity of IMC is particularly attractive for studies at low levels of stressors, where lethality-based data are inadequate. We have revealed via IMC the effect of low dose rates from radioactive β^−^-decay on bacterial metabolism. The low dose rate regime (<400 µGyh^−1^) is typical of radioactively contaminated environmental sites, where chemical toxicity and radioactivity-mediated effects coexist without a predominance or specific characteristic of either of them. We found that IMC allows distinguishing the two sources of metabolic interference on the basis of “isotope-editing” and advanced thermogram analyses. The stable and radioactive europium isotopes ^153^Eu and ^152^Eu, respectively, were employed in monitoring *Lactococcus lactis* cultures via IMC. β^−^-emission (electrons) was found to increase initial culture growth by increased nutrient uptake efficiency, which compensates for a reduced maximal cell division rate. Direct adsorption of the radionuclide to the biomass, revealed by mass spectrometry, is critical for both the initial stress response and the “dilution” of radioactivity-mediated damage at later culture stages, which are dominated by the chemical toxicity of Eu.

## 1. Introduction

Heavy metal toxicity in microorganisms is a well-known phenomenon that originates in the coordination and redox chemistry of predominantly transition metals. Their complexes with intracellular biomolecules affect redox potentials and/or prevent other essential metal ions from binding to the specific sites required for the catalytic function of enzymes. Consequently, metal toxicity may be measured, for example, by altered enzyme activities [1]. On the other hand, adaptive strategies have been demonstrated in a variety of microorganisms and in complex ecological systems [2] revealing graded sensitivities to metal ions [3]. The metabolism of such microorganisms may be exploited for microbial bioremediation of heavy-metal-contaminated sites [4].

Assessment of toxicity is particularly complicated for radionuclides because in addition to their chemical toxicity, radical formation is observed as a chemical consequence of the various types of ionizing radiation emitted during radioactive decay. Environmental studies have mostly focused on the interaction of natural uranium with microorganisms and plants, where toxicity is largely a consequence of its chemotoxicity due to its coordination and redox chemistry as reviewed [5]. The actual effects of gamma radiation and alpha particles on microorganisms and tissues have been studied extensively in vitro using external radiation regimes that revealed that DNA double strand breaks correlate well with lethality [6], exhibiting a relative biological effectiveness typically between one and three for electrons, protons, α-particles and light ions. [7,8,9].

The effects of ionizing radiation on microorganisms, including complex bacterial communities, are typically derived from experiments with external radiation sources providing helium ions, gamma rays or electron beams, [9,10] followed by post-exposure chemical and genetic analyses of the biological materials. However, the actual radiation dose rate encountered by a microorganism in a radionuclide-containing environment depends critically also on the chemical properties of the specific radionuclides, such as their association with cell wall components, intracellular accumulations or active cellular export strategies. Furthermore, the majority of the published work addresses lethality to derive measures of radiotoxicity in the form of radiation doses required for a certain percentage of survivor cells or minimum doses required for efficient bactericidal effects.

The experiments presented here aim at elucidating whether isothermal microcalorimetry (IMC) enables recording the impact of low-dose radiation from ^152^Eu on bacterial metabolism using *Lactococcus lactis* as a previously established model organism [11]. The measurement of metabolic thermal power via IMC appears particularly appealing in the context of low-dose radiotoxicity assessments for the following reasons: In contrast to external radiation exposure, the chemical interactions of the heavy metals with the studied microorganism are preserved in IMC and correspond to a realistic environmental situation. This avoids the uncertainties in extrapolating lethality-based data to low-dose toxicity and prevents inconsistencies in the concentration-dependent speciation of radionuclides [12]. The unprecedented sensitivity of state-of-the-art isothermal IMC appears appropriate to detect even subtle effects expected at low doses, provided that appropriate evaluation methods are established. Finally, safety regulations render thorough biochemical analyses difficult if not impossible with radioactive elements but can be adhered to in calorimetric experiments, which employ sealed ampoules only. Such ampoules can be autoclaved and discarded without ever being opened again, minimizing human exposure risks.

As an environmentally more realistic experiment than external radiation studies, IMC detects both chemical and radiation-induced toxicity of a radionuclide. A particular challenge is the distinction between both. We show here that the use of radionuclide isotopes provides a powerful strategy to separate these effects. However, the analysis of IMC traces at the required degree of detail requires a methodology that parametrizes the obtained thermograms to express toxicity quantitatively and, where necessary, in a growth-phase-dependent manner. Such an analysis tool was developed recently [13]. Here, it is used to reveal the impact of β-radiation from ^152^Eu on bacterial growth in comparison to that of the non-radioactive ^153^Eu isotope (used in the following as short notation for the “natural Eu” isotope mixture dominated by this isotope). Typically, secondary electrons are produced stochastically by ionizing radiation, leading to radical formation and hydrated electrons as endpoints of the chemical phase of radiation damage in cells [14]. In the specific case of beta-minus decay, electrons are the primary interacting particles already and their influence on metabolism is analyzed here. The results demonstrate that advanced evaluation methods of IMC are crucial to fully exploit the potential of this technique in the non-invasive detection of toxicity in living cultured microorganisms.

## 2. Materials and Methods

### 2.1. Lactococcus lactis Cell Culture

*Lactococcus lactis* (*L. lactis*) subsp. cremoris strain NZ9000 was obtained from Nizo Food Research BV, Ede, The Netherlands. Bacteria were grown at pH 7.2 in M17 medium containing (per liter) 5 g Neopeptone, 5 g Bacto Soyton, 5 g beef extract, 2.5 g yeast extract, 0.5 g ascorbic acid, and 1 g Disodium-2β-glycerophosphate · 5H_2_O. The mixture was autoclaved and supplemented with 1 mL of 1 M MgSO_4_ and 20 mL of 50% glucose, both sterile-filtered.

### 2.2. Isothermal Microcalorimetry of L. lactis Cultures

In order to test growth response to the presence of stable ^153^Eu, microcalorimetric ampoules were filled with 4 mL M17 medium to exclude air space to ensure anaerobic metabolism and supplemented with aliquots of a sterile-filtered EuCl_3_ solution (10 mM), reaching final concentrations of 50 to 500 µM Eu. For microcalorimetric experiments with the radioactive isotope ^152^Eu at a constant total background concentration of 100 µM EuCl_3_, 20 to 200 µL of a sterile-filtered ^152^Eu-containing Eu-stock solution (in 100 mM HCl) was added shortly before the experiment. The activity of the ^152^Eu stock solution was 25.2 kBq per mL corresponding to 0.0258% ^152^Eu in a total of 100 µM EuCl_3_. Finally, the samples were inoculated with 20 µL of *L. lactis* culture (diluted with medium to an optical density at 600 nm (OD600) of 0.1). This corresponds to an initial number of 100,000 cfu (±40%) [15]. Residual oxygen in the medium at the start of the experiment was rapidly consumed, resulting in anaerobic, fermentative growth [16]. Isothermal microcalorimetry was performed with a TAMIII instrument (Waters GmbH, Eschborn, Germany) equipped with 12 channels to continuously monitor the metabolic heat flow produced by the bacterial cultures at 20 °C for up to 80 h.

### 2.3. Brassica Napus Cell Culture

In order to demonstrate the generality of “Isotope-Editing” of metabolic heat data, additional experiments were carried out with cultured plant cells in the presence of natural uranium and the α-emitter ^233^U (Appendix A). *Brassica napus* (*B. napus*) suspension cells were cultivated in liquid modified Linsmaier and Skoog medium (medium R, [17]) starting from callus cells (PC-1113; DSMZ, Braunschweig, Germany) as described [18]. 

Every 7 days, cells were subcultured into fresh culture medium to maintain the cell culture. For uranium exposure experiments, 20 mL cell suspensions were filtered through a nylon mesh (50 µm pore size; Bückmann GmbH, Mönchengladbach, Germany) without suction. Subsequently, cells were rinsed with medium R with a reduced phosphate concentration of 6.25 µM representing 0.5% of the original phosphate concentration (medium R_red_) [19]. This medium was used to limit the precipitation of hardly soluble uranyl (VI) phosphate complexes.

### 2.4. Isothermal Microcalorimetry of B. napus Cells in the Presence of ^233^U

For microcalorimetric experiments, 0.3 g of wet cells were transferred into 4 mL ampules. To differentiate between the radiotoxic and chemotoxic effects of uranium, *B. napus* cells were simultaneously exposed to natural uranium (U_nat_) as well as the uranium isotope U-233 (^233^U). The U_nat_ concentration in medium R_red_ was varied between 50 µM and 35 µM and that of ^233^U between 0 and 15 µM in parallel. In each case, a total uranium concentration of 50 µM (U_nat_ + ^233^U) was obtained. Accordingly, aliquots of sterile aqueous UO_2_(NO_3_)_2_ (1.11 × 10^−2^ M, 9.45 × 10^−3^ M, or 9.83 × 10^−3^ M; ~60 Bq/mL) and ^233^UO_2_(NO_3_)_2_ (1.02 × 10^−3^ M; 8.45 × 10^4^ Bq/mL) stock solutions were added to medium R_red_. The pH values of the solutions were adjusted to pH 5.8 ± 0.1 (pH meter pH720, WTW inolab, Weilheim, Germany, Blue Line 16 pH electrode, SI Analytics Mainz, Germany). Subsequently, 2 mL of the respective medium was added to the cells and the ampules were tightly capped. For each experiment, control samples of medium R_red_ as well as of cells in medium R_red_ without uranium were prepared. The background concentration of 50 µM of natural uranium is less than the half-maximal inhibitory concentration of U(VI) for *B. napus* estimated by oxidoreductase activity and IMC [18] and is thus below the minimal inhibitory concentration (MIC).

Before measurement, the samples were held in a TAM III in a waiting position for 15 min before complete insertion followed by 45 min of equilibration. In each experiment, samples of the same U_nat_ or ^233^U concentration were recorded at least in duplicate. After the microcalorimetric measurement, the bioassociated amount of uranium was determined using inductively coupled plasma mass spectrometry (ICP-MS; models NexION 350x, Perkin Elmer, Rodgau, Germany and iCapRQ, Thermo Fisher Scientific, Dreieich, Germany) for selected samples containing U_nat_ only or via liquid scintillation counting (LSC; model 300 SL, Hidex, Mainz, Germany) for ^233^U containing samples.

## 3. Results

Isothermal microcalorimetric experiments were performed to detect the influence of the heavy metal europium on the metabolic activity of cultures of the Gram-negative bacterium *L. lactis*, a previously studied model organism for assessing heavy metal toxicity [11]. Here, the effects of two europium isotopes are compared for the purpose of distinguishing radiation-mediated toxicity from chemical toxicity. The non-radioactive isotope ^153^Eu was used to quantitate the chemical toxicity of the lanthanide. Figure 1 shows the time-course of the metabolic heat release from a culture grown in the absence of 100 µM ^153^Eu. Two metabolic phases are discernible by their peak metabolic activities occurring after ~12 h and ~15 h of growth, and a third late phase gives rise to a broad shoulder after ~16 h of growth before metabolic activity declined rapidly. The succession and thermal activity of these metabolic phases were barely affected by 100 µM of the stable isotope ^153^Eu. The only visibly discernable features are the slightly deeper gap between the first and second metabolic phase in the control (arrows in Figure 1) and the lower peak metabolic activity in the presence of 100 µM Eu. The very minor changes induced by the lanthanide show that the minimal inhibitory concentration (MIC) of the stable Eu isotope is surely larger than 100 µM. However, the time dependence of the metabolic thermal power *P(t)* is rather complicated and the presence of substructures in the thermograms is at odds with simulations using classical growth models such as Richards- or Gompertz functions [13]. Similar to many other growth models, the latter two exhibit only a single inflection point (the maximum of the first derivative, i.e., the maximal absolute growth rate). Since the IMC traces scale with the first derivative of the bacterial growth curve—because metabolism is strictly coupled to culture growth—classical growth models would predict only a single thermogram peak. The clearly resolved thermogram substructures indicate the presence of multiple inflection points in the time-dependent bacterial growth curves. Therefore, we analyzed the thermograms in dependence of the total released heat (enthalpy *H*), rather than as a function of time (as displayed in Figure 1). This results in simpler thermogram shapes (Figure 2) and the released heat (plotted along the x-axis) scales directly with consumed nutrients. Thus, the nutrient-dependence of growth is directly visible. Simple and also multiple successive nutrient–metabolism relations can be formulated analytically without requiring an analytical solution in the time domain. In the enthalpy *(H)* domain, the thermal power *P(H)* could be fitted extraordinarily well with the empirically derived function:
Figure 1Thermograms of *L. lactis* cultures in the absence (**A**) and presence (**B**) of 100 µM ^153^Eu. Simulated thermograms (dotted lines) were fitted according to Equations (1) and (2) using the analysis method dAR-TS [13]. Details are described in the text. The metabolic activity exhibited an initial fast exponential phase (black dotted line) followed by two consecutive metabolic states with slower growth (red and green dotted lines) evident from their extrapolated initial exponential growth phases (ca. first 10 h). The simulated curves cover both rising and falling metabolic activities. The presence of the stable ^153^Eu had little effect on the thermogram shapes and intensities (dashed vertical lines mark the maximal thermal power for each metabolic phase in the absence of Eu). Arrows indicate small visible differences in time and magnitude of the maximal metabolic activity reached by the first metabolic phase. The root mean square deviation (rmsd) of the data from the three fitted data ranges is less than 1% of the peak maximum in each of the respective metabolic phases.
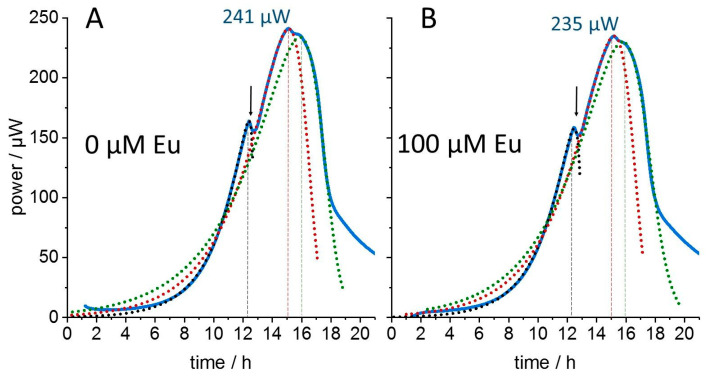

Figure 2Enthalpy plots of thermograms from *L. lactis* cultures. The medium without Eu (**A**) exhibited an initial growth rate *r_i_* = 0.47 h^−1^. (**B**) Culture supplemented with 100 µM ^153^Eu (*r_i_* = 0.43 h^−1^) serving as reference for the experiments with ^152^Eu. The same data are displayed as in Figure 1. (**C**) and (**D**) show the metabolic activity in the presence of 150 and 300 µM ^153^Eu, respectively, accompanied by a further decrease in the peak metabolic power. Shaded areas mark the data ranges that were used for the thermogram simulation via dAR-TS for each metabolic phase (first, second, third phase in gray, red, green, respectively; dotted lines show the full thermogram simulation for completeness). The error range of initial growth rates is ±8% when the fitted curve is allowed to adopt a root mean square deviation of ±2% from the raw data. The displayed fits have a rmsd of less than 1% from the raw data within the fitted shaded areas.
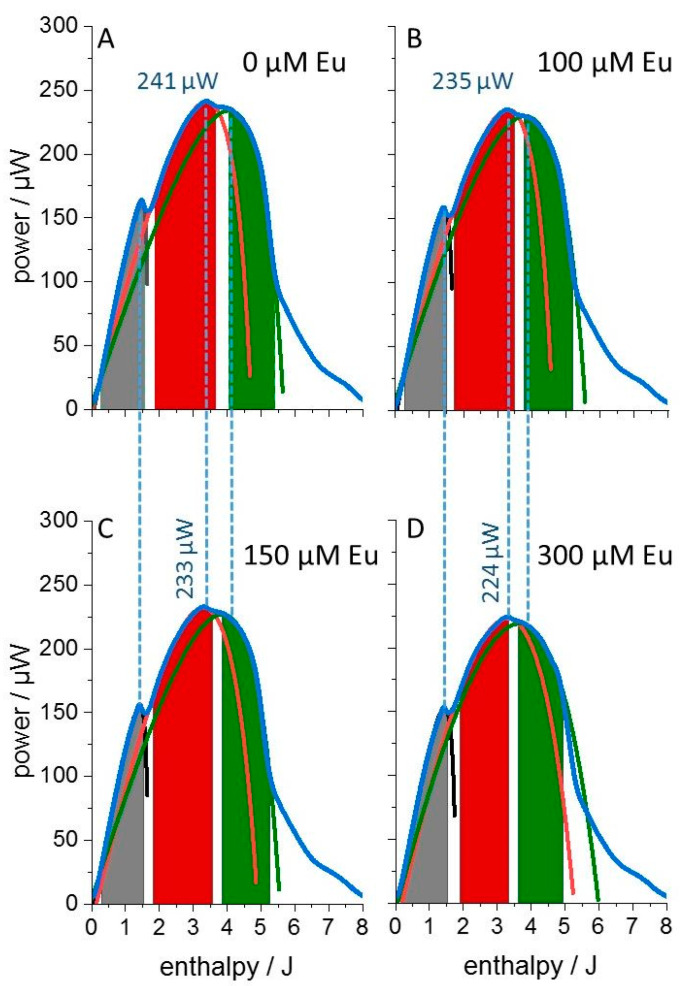

(1)$$P\left(H\right)=H\cdot r_{0}\cdot \theta \left(H\right)\;with\;H\left(T\right)=\int_{0}^{T}P\left(t\right)dt\;and\;\theta \left(H\right):=\left(H_{0}-H\right)/\left(H_{0}-H+H_{f}\right)$$
(2)$$t=\int_{0}^{H}\left[1/P\left(h\right)\right]dh$$
with *h* the enthalpy as the integration variable, as opposed to *H*, the upper integral boundary, i.e., the total enthalpy released up to the time *t*. The fitted curves in Figure 1 were obtained by transforming the data for *P(H)* from the enthalpy domain back to the time domain *P(t)* using Equation (2). The description of metabolic activity by Equation (1) includes the “metabolic load” (0 < *Θ* < 1), which expresses in a Langmuir-adsorption-like hyperbolic manner the degree to which bacterial metabolism is saturated by nutrients. Nutrient adsorption is followed by metabolic turnover, which generates thermal power *P*, increases the number of nutrient-adsorbing bacteria (in proportion to *H*) and decreases the concentration of nutrients (in proportion to *H*_0_ − *H*). In fact, the equation for *Θ* exhibits the same mathematical structure as the Michaelis–Menten equation for the rate of enzymatic activity as a function of substrate, where the substrate concentration is replaced here by the term *H*_0_ − *H*. The latter calorimetric quantity scales with the amount of nutrient during a calorimetric experiment, because *H*_0_ stands for the maximal heat release at the end of a metabolic phase, i.e., the total released heat upon consumption of all nutrients in a given specific metabolic pathway) and *H* is the heat released up to any given time during the metabolic activity. Mathematically, *H_f_* corresponds to the Michaelis–Menten (or Monod) constant and *r*_0_ to the maximal catalytic rate of an enzyme (or the maximal growth rate of a bacterial culture in the Monod formalism). Thus, *Θ(H)* also expresses a hyperbolic relation between *P* and *H*. The parameters *H*_0_, *r*_0_ and *H_f_* were fitted to reproduce the experimental data according to Equation (1) as described [13]. Using this “dynamic adsorption reaction thermogram simulation” (dAR-TS), each peak (or shoulder) in the thermograms could be reproduced with the optimized parameters *r*_0_: maximal growth rate of the bacteria at “metabolic load” *Θ* = 1; *H*_0_: calorimetric equivalent of the total initial amount of nutrients; *H_f_*: calorimetric equivalent of the amount of nutrient supporting half-maximal initial growth. The analysis was described in detail in an accompanying paper [13]. In this way, the individual time-dependent heat flow contributions of each metabolic phase to the complex time-dependent thermograms in Figure 1 were obtained with high accuracy using consecutive (per metabolic phase) curve fitting. The fitted curves reproduce the maximally possible number of original data points fulfilling the criterion of an rsmd smaller than 1% of the experimentally measured peak heat flow. The corresponding data intervals are indicated in Figure 2A,B (colored data ranges), which show the original data as a function of the released heat *H*. The full traces outside the selected data ranges in Figure 1 and Figure 2 are shown only for completeness and visualizing the extrapolated initial exponential growth phases. In fact, the modeled “inverted parabola-like” traces show an initial slope equivalent to that of a logarithmic plot of the metabolic heat flow data versus time, which represents the initial exponential growth rate *r_i_* of a culture given as:(3)$$r_{i}=r_{0}\cdot \theta \left(0\right)$$

As *Θ*(0) equals *H*_0_/(*H*_0_ + *H_f_*), the relation between *r*_0_ and *r_i_* is of the same formal structure as that for the initial growth rate *r_i_* of a bacterial culture, where *r_i_* equals *r_max_∙S*_0_/(*S*_0_ + *C*), with the initial substrate concentration *S*_0_ (analog to *H*_0_) and the Monod constant *C* (analog to *H_f_*). Analysis using dAR-TS allows extrapolating such initial rates to each of the discernible metabolic phases (characterized by their local peaks) additionally to the experimentally observed initial exponential growth of the culture. Accordingly, linear sections in the enthalpy plot near *H* = 0 are discernible for each metabolic phase. However, it is important to note that it is exclusively the curvature in the original thermograms within the selected data ranges (where fit and raw data are indistinguishable by the eye) from which the entire fits were obtained without forcing the curves through the origin. In the same manner, thermograms recorded up to 300 µM ^153^Eu were analyzed and revealed that culture growth was very little affected by the lanthanide (Figure 2C,D).

The sketched thermogram simulation via dAR-TS allowed us to detect subtle differences between the otherwise hard-to-distinguish thermograms. The goal of the following extended data analysis is the determination of the three thermogram parameters *r*_0_, *H*_0_ and *H_f_* for each growth phase. Based on these numbers, the distinction between chemical sensitivity and radiation sensitivity of the bacterial culture is attempted by comparing these parameters for cultures grown in media supplemented with either the stable ^153^Eu- or the radioactive ^152^Eu- isotope at an identical total lanthanide concentration of 100 µM. Figure 3 shows the enthalpy plots of cultures grown in the presence of 250 pM and 1 nM ^152^Eu. All three metabolic phases were again observed. The early peak in heat release after about 12 h of growth (corresponding to 1–2 Joule of total heat release) was reduced at the lower concentration and the switch from the first to the second metabolic phase occurred earlier (and at lower total heat release).

The more subtle effects of the radioisotope on the thermogram shape were analyzed according to Equation (1) to reveal the effects on both the initial growth rate r_i_ and the apparent nutrient affinity expressed by the “metabolic load” *Θ*. The dependence of these two parameters on the EuCl_3_ concentration is shown in the “rate plot” in Figure 4A for the bacterial growth in the presence of only the stable isotope ^153^Eu at concentrations between 0 and 300 µM. The data were scaled to a metabolic load of *Θ* = 0.5 and a maximal bacterial growth rate *r*_0_ = 1 in the reference culture without the lanthanide (thermogram from Figure 1A). This results in the reference point being in the center of the rate plot with *r_i_* = 0.5 and *Θ* = 0.5 as predicted by Equation (3). Based on the same Equation, a toxic effect exclusively on nutrient uptake reduces the “metabolic load” *Θ* but not the maximal growth rate *r*_0_. The initial growth rate *r_i_*, on the other hand, does decrease to the same extent as *Θ* as evident from Equation (3). Thereby, toxicity on exclusively the “metabolic load” causes a diagonal displacement in the “rate plot” relative to the reference point. This form of toxicity was found for the stable isotope, which affected the first metabolic phase of the bacterial cultures with almost mathematical exactitude: the data pairs for *r_i_* and *Θ* were displaced strictly diagonally upon increasing the ^153^Eu concentration. Consequently, the stable isotope exclusively affected the “metabolic load” (and in consequence the initial growth rate) during the first metabolic phase but did not change the maximally possible growth rate *r*_0_ of the bacteria. Similar trends were observed for the second and third metabolic phases for which *r_i_* and *Θ* decreased with increasing Eu concentration. However, in both metabolic phases, a larger initial growth rate *r_i_* was obtained than expected for an exclusive effect on the “metabolic load”. In these metabolic phases, the bacteria reacted also with an increase of their maximal growth rate *r*_0_. Thereby, initial growth rates *r_i_* increased in the same proportion as *r*_0_, which leads to [*Θ*, *r_i_*] data pairs that lie above the diagonal of the “rate plot”.

An equivalent analysis was performed with the radioactive isotope ^152^Eu (in the pM to nM range) at a total background concentration of the lanthanide of 100 µM. Therefore, the reference point for the “rate plot” in Figure 4B was obtained from the growth of a culture in the presence of 100 µM ^153^Eu, i.e., in order to provide a chemically identical but non-radioactively doped medium. The “rate plot” shows that the first metabolic phase was most sensitively affected by the introduced radioactivity. The “metabolic load” clearly increased in the presence of the radioisotope, as the data points are clustered to the right of the reference point. A corresponding diagonal displacement to also higher initial growth rates is visible only at the smallest ^152^Eu concentration. However, the data cluster below the diagonal, indicating the reduction of the maximal growth rate *r*_0_. The downward vertical displacement is shown for the data pair representing the highest ^153^Eu concentration (arrow). The second and third metabolic phases responded to the radioisotope very differently from the initial culture phase. In the second metabolic phase, metabolic load and initial growth rate decreased such that the “rate plot” displays again a roughly diagonal displacement to the left, indicative of a lowered apparent nutrient affinity. In contrast, the third metabolic phase was almost unaffected by the radioisotope.

This ensemble of data confirms previous observations that metal toxicity comprises non-uniform metabolic responses of microorganisms related to different metabolic phases indicative of different enzyme repertoires and biochemical pathways with distinct metal sensitivities [11]. Whereas growth-phase-specific chemical toxicity is biochemically plausible, growth-phase-specific radiation-sensitivity to the radionuclide ^152^Eu is not intuitive, because radiation damage is stochastic and independent of the specific metal biomolecule interactions involved in chemical toxicity.

## 4. Discussion

We found that chemical and radiation-mediated toxicity of heavy metals for bacterial metabolism can be distinguished using microcalorimetric measurements. The key to this distinction is a thorough data analysis performed here using a “dynamic adsorption reaction thermogram simulation” (dAR-TS) [13]. The strength of the analysis lies in the dual-parameter description of toxicity (initial growth rate *r_i_* and “metabolic load” *Θ*), which revealed the growth-phase-dependence of both the chemical toxicity of Eu and the sensitivity to sub-lethal effects of β^+^ and β^−^ radiation from ^152^Eu decay. Chemical toxicity led to a decrease in metabolic load in all three metabolic phases of the growth of *L. lactis* cultures between 0 and 300 µM ^153^Eu, evident from the diagonal displacement of *r_i_*, *Θ* data pairs relative to the reference without Eu. The slight positive deviation from the diagonal in the second and third metabolic phase indicates that the cultures compensated for the lowered nutrient affinity with an enhanced maximal division rate *r*_0_. Thereby, the net culture growth was little affected, as expected from the similarity in the “enthalpy plots” in Figure 1.

In contrast, a predominant effect of β-radiation on the first metabolic phase was revealed by the rate plot in Figure 4B, showing an unexpected enhancement of the “metabolic load”, i.e., an enhanced uptake of nutrients. The toxic effect of β-radiation is thus opposite to that induced chemically by increasing the metal concentration. Likewise, the adaptational response differs from that to chemical toxicity: the maximal growth rate *r*_0_ is reduced up to 50% (arrow in the double logarithmic “rate plot” in Figure 4B). This reduction is compensated by the increased metabolic load, such that all data pairs of the first metabolic phase cluster at *r_i_* values that are similar to or slightly larger than the reference. It appears that the bacteria cope with the electron-mediated damage by enhancing the nutrient uptake efficiency, thereby “diluting out” the radioactive element by enhanced initial growth (in line with a steeper and more prolonged first metabolic phase in the enthalpy plot in Figure 3B as compared to 3A). Such an adaptive response to radio-sensitivity would be efficient if the radioisotope associates with the bacteria externally or by intracellular uptake. In both cases, enhanced growth reduces the Eu load per cell. We found indeed that Eu was strongly associated with the biomass using ICP-MS such that after 15 h of growth, more than 90% of the metal was found in the pellet after centrifugation of the bacteria (Appendix A). An adaptive stress response in the form of enhanced initial growth is further supported by the fact that ^152^Eu affected the first metabolic phase the most, the second partially and the third phase not at all. The attenuation of radiation sensitivity during growth agrees with partitioning of the radioisotope among an increasing number of cells, such that the bacteria suffer from radiation damage much less at late culture phase. Furthermore, there is no indication that the radionuclide caused partial cell death in the initial inoculate (causing a radioisotope-dependent lag phase before exponential growth became measurable). Thus, the experiment appears to correctly simulate non-lethal low dose rate radiation effects, which prevail in most environmental conditions and are the most difficult to detect.

Realistic estimates of radionuclide toxicity at low dose rates are in fact required in fundamental research on the mechanisms of radiation damage in organisms as well as for the definition of concentration thresholds for environmental safety regulations. In the case of heavy metals, such data should not be derived from lethality data, because lethality requires much higher metal or radionuclide concentrations than those found in the environmentally relevant low dose rate regime. Cells and organisms survive and reproduce at low dose rates but nevertheless react to chemical and radiation-mediated toxicity. The major concern regarding lethality-based toxicity measures originates in the complicated and diverse speciation of radionuclides, which changes significantly with radionuclide concentration, thereby affecting bioavailability and interference with biochemistry particularly for actinides [12]. Importantly, the approach chosen here identifies the radiotoxic component separately from the chemical toxicity of the heavy metal via isotope-specific measurements of metabolic heat release using the stable ^153^Eu and the radioactive ^152^Eu isotopes. This is an important improvement compared to studies using external radiation sources, thereby neglecting the normally accompanying chemical toxicity. Prominent functional groups of biomolecules acting as complexing agents for f-block metals have been surveyed, revealing a plethora of possible chemical interferences with metabolism [20] and substantiating the strong bioassociation of Eu found in the experiments described here. Additionally, the thermodynamics of europium speciation in the presence of inorganic anions has been reviewed, [21] which underlines the need to avoid extrapolations from high to low metal concentrations.

The present study specifically addresses the potential of IMC for detecting effects of low dose rates on microbial metabolism. ^152^Eu decays with 72.1% to ^152^Sm (β^+^-decay almost exclusively via electron capture, total energy Q^+^(^152^E) = 1874.3 keV and average radiation energy of 1345 keV) and 27.9% to ^152^Gd (β^−^-decay, Q^-^(^152^E) = 1818.8 keV and average radiation energy 508 keV) [22]. The total dose rate released in an ampoule volume of 4 mL at the highest ^152^Eu concentration (1.23 nM with 1.2 kBq mL^−1^) corresponded to 1.29 mGyh^−1^ of which 60% was estimated to be in the form of γ-radiation [22]. The accompanying gamma energies lie between 0.1 and 1.8 MeV, where attenuation coefficients of water fall in the range of 4.9 × 10^−2^ to 1.7 × 10^−1^ cm^2^g^−1^ [23]. With an ampoule diameter of 1 cm and filling of 4 mL, we estimate that about 80% of the γ-radiation exited the growth medium, which led to an estimated maximal dose rate of ca. 600 µGyh^−1^ within the ampoule. Even if the escape of γ-radiation is neglected, the experiments clearly cover the low dose rate regime defined here as <400 µGyh^−1^ [24] and the observed metabolic responses can be largely assigned to the effect of electron-mediated processes (the formation of Gd and Sm decay products is vanishingly small: still less than nM concentration after the ^152^Eu half-life of more than 13.5 years).

It is interesting to relate the data to previous IMC data from *L. lactis* that determined a heat production of ~8 nJ per formation of one cfu [15]. This implies that about 10^8^ to 10^9^ cells accumulated in the third metabolic phase. Even with a prolonged life cycle of ~5 h in the late culture (as compared to ~2 h initially, Figure 1), about one decay process would on average occur per cellular life cycle in the presence of 1.23 nM ^152^Eu if all of the lanthanide associated with biomass. Such a low exposure agrees with an essentially unaffected metabolism at late culture stages. The dose rate per cell cycle would have been two to three orders of magnitude higher if the same total amount of ^152^Eu associated with the initial inoculate (a few 100,000 cells). It is remarkable that IMC detects metabolic responses to radioactive decay despite the sublethal dose rates present at any stage of the culture growth. 

Whereas we emphasize the strength of thorough IMC data analysis in the context of sublethal radiation damage, it is beyond the scope of this work to evaluate the underlying biochemical pathways. However, at the low dose rates applied here, oxidative stress is not likely to play a role in the toxicity of β-radiation [24]. The relation between the relative biological effectiveness (RBE) and the linear energy transfer (LET) for different types of ionizing radiation has been explored and found to be diverse depending on the cluster size of primary radiation damage [25] as reviewed [26]. In the case of ^152^Eu decay, the maximal electron energy of 695.6 keV corresponds to a mean free path (in water) of elastic and inelastic scattering of about 2 nm and 12 nm, respectively [27]. Thus, cell-surface-adsorbed ^152^Eu would already start depositing part of its energy during its early trajectory in the direct vicinity of the bacterial cell (more so with electrons emitted at lower energies than the predominant maximal energy). Cellular ionization events as a consequence of the direct chemical adsorption of ^152^Eu to the cell surface appear to be the prime cause for the observed metabolic effects of β^−^-radiation and would be impossible to mimic with an external source.

The use of different isotopes was essential in this study. Similar “isotope-edited” studies can in principle be performed with other isotopes and cell types. For example, the effect of the strong α-emitter ^233^U on plant cell metabolism can be monitored on the background of natural uranium (Appendix A). Direct adsorption to the biomass has been observed for the plant cell culture as well (Appendix A). However, in the case of bacterial growth, the effect of α-radition on metabolism complies with an initial all or nothing mechanism, reducing the viable inoculate size but barely affecting the initial growth of the survivors (Appendix A).

## 5. Conclusions

The advancement of systematic quantitative analysis tools is a critical step in extracting high information content from microbial thermograms. We demonstrate that a “dynamic adsorption reaction thermogram simulation” (dAR-TS) reveals the multidimensionality of toxicity in the form of combined effects on maximal division rate and nutrient affinity. The sensitivity of dAR-TS was challenged here in the particularly demanding “isotope-edited” approach to the distinction of chemical toxicity from radiation-mediated toxicity. Based on these and previous data evaluations, IMC analysis methods will be further developed into a web-tool for the evaluation of IMC data with the aim of promoting research on the energetics of microbial growth and increasing the comparability of IMC data between labs.

## Figures and Tables

**Figure 3 microorganisms-11-00584-f003:**
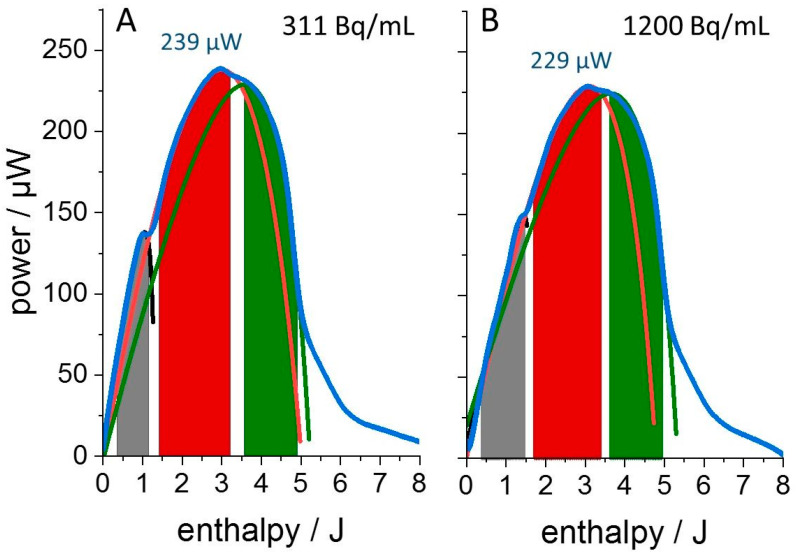
“Enthalpy plots” of thermograms from *L. lactis* cultures grown in the presence of the radioactive isotope ^152^Eu. The total concentration of EuCl_3_ in the growth medium was 100 µM supplemented with (**A**) 320 pM ^152^Eu, (311 Bq/mL) or (**B**) 1230 pM ^152^Eu (1200 Bq/mL). See Appendix A for more thermograms recorded in the presence of ^152^Eu. Concentrations are rounded to the closest multiple of 10 pM. Color code as in legend to Figure 2.

**Figure 4 microorganisms-11-00584-f004:**
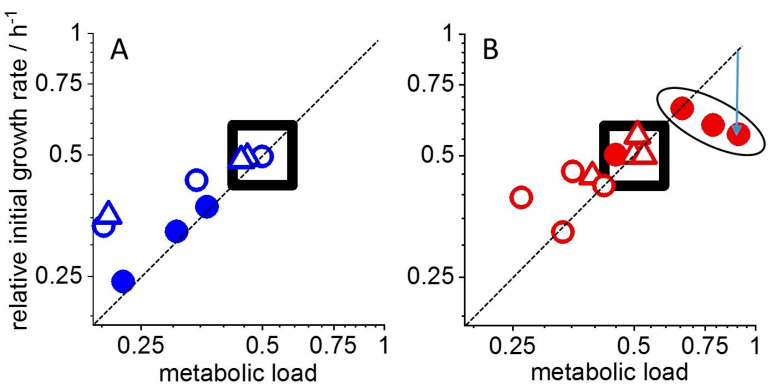
”Rate plot” of bacterial growth parameters. The initial growth rate *r_i_* is plotted against the “metabolic load” for the first (filled circles), second (open circles) and third growth phases (triangles). (**A**) “Rate plot” for concentrations of 0, 100, 150 and 300 µM ^153^Eu (data from Figure 2, blue symbols). (**B**) “Rate plot” for concentrations of 0, 130, 320, 630 and 1230 pM ^152^Eu (rounded to closest multiple of 10 pM; Figure 3 and Appendix A, red symbols) in the presence of a background concentration of 100 µM ^153^Eu. All data were scaled to an initial metabolic load of 0.5 and a maximal growth rate *r*_0_ of unity leading to the reference point [*0.5*; *0.5*] (marked by square boxes) as described [13]. The diagonal (dashed line) corresponds to the change of the relative initial growth rate *r_i_* if only the metabolic load was affected by the different Eu concentrations. Deviations from the diagonal indicate an additional increase or decrease of the maximal growth rate *r*_0_ for data points lying above or below the diagonal, respectively, as follows from Equation (3).

## Data Availability

Data from this study are available on request from the author.

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
