# Peer review of "Distinct Effects of Chemical Toxicity and Radioactivity on Metabolic Heat of Cultured Cells Revealed by “Isotope-Editing”"

_microorganisms, 2023, doi:10.3390/microorganisms11030584_

Round 1

Reviewer 1 Report

Dear authors,

Congratulations on this excellent approach, which allows to distinguish between two mechanisms of radio nucleotide toxicity. Nevertheless, I think your manuscript needs and deserves a thorough revision so that microbiologists with an average physical and mathematical education can follow your thoughts.

My main concerns:

- The description of data evaluation is hardly understandable at the current level of the manuscript without intensive reading of the pre-work (Fahmy Microorganisms 2022,10). Maybe you can improve it already by:

1.)         Adding a list of symbols with units

2.)         Take attention to the units in the figures (e.g. growth rate surely has the unit h-1 in Fig. 4)

3.)         Please explain better how the initial growth rate was calculated as a function of the metabolic load.

4.)         Please improve the quality of figures 2-3. The dotted sub-functions are hard to see.

- 153Eu decays into Gd and Sm. Wouldn't the toxicity of the decay products have to be taken into account, or is the concentration or the decay rate simply too small? I think this point deserve a clarifying sentence.

Minor points:

Maybe just a matter of taste: The terms "thermogram, thermal power" are unusual. The typical terms in literature are heat flow, heat flux or heat production rate.

L 19: Delete „-„

L 39: Don't the decay products also have an effect on the overall toxicity?

L 46: A comma is missing between alpha particles and protons.

L67: Delete ii)

L70: Delete iii)

L95: H deep 2 O

L98: “4 mL”. You should explain why you have chosen this amount. It is probably to exclude an air space with the respective oxygen diffusion. To ensure anaerobic metabolism.

L 122: A space is missing.

L124: “4 mL” for the experiments with plant cells? Typically, plant cells needs oxygen for growth.

Section 2.2: It is not clear to me what the experiments with the plant cells are needed for, which are barely discussed in the main article. I would omit the section.

L186: Eq. (2): The variable “h” is not explained.

L210: Legend to Figure 2: Please provide the error range of the different initial growth rates.

L223: Omit “)”

L270, Fig. 4: Please provide the unit for the y-axis.

L 367: “6 nJ s-1 “ please add the data in W (6 nW) because the calorimeter measure in W.

Supplementary Material: Unfortunately, I got an error message (Error 404 – File not found) and could therefore not review the SI.

Author Response

RESPONSE TO REVIEWER 1

We thank the reviewer for the appreciation of our work and for very helpful comments to which we respond (resp.:) point by point below in the original text provided by the referee.

The requested revisions have been carried out and have helped improving the manuscript substantially.

REVIEWER 1

Open Review

English language and style

( ) English very difficult to understand/incomprehensible
( ) Extensive editing of English language and style required
( ) Moderate English changes required
( ) English language and style are fine/minor spell check required
(x) I don't feel qualified to judge about the English language and style

Yes

Can be improved

Must be improved

Not applicable

Does the introduction provide sufficient background and include all relevant references?

(x)

( )

( )

( )

Are all the cited references relevant to the research?

(x)

( )

( )

( )

Is the research design appropriate?

(x)

( )

( )

( )

Are the methods adequately described?

( )

( )

(x)

( )

Are the results clearly presented?

( )

( )

(x)

( )

Are the conclusions supported by the results?

(x)

( )

( )

( )

Dear authors,

Congratulations on this excellent approach, which allows to distinguish between two mechanisms of radio nucleotide toxicity. Nevertheless, I think your manuscript needs and deserves a thorough revision so that microbiologists with an average physical and mathematical education can follow your thoughts.

My main concerns:

- The description of data evaluation is hardly understandable at the current level of the manuscript without intensive reading of the pre-work (Fahmy Microorganisms 2022,10). Maybe you can improve it already by:

1.)        Adding a list of symbols with units

resp.:   a list of mathematical symbols has been added, including units (lines 151-160)

2.)        Take attention to the units in the figures (e.g. growth rate surely has the unit h-1 in Fig. 4)

resp.:     Figure 4 shows the normalized initial growth rate, i.e., for conditions at which the control exhibits a growth rate of 0.5 h-1.This is now more clearly expressed by the y-axis label (to which the unit has been added as well) and it is stated in the figure legend.

3.)        Please explain better how the initial growth rate was calculated as a function of the metabolic load.

resp.:   In lines 224-235 and 268-271 we describe in more detail that the relation between heat flow (metabolic activity) and total released heat (enthalpy) follows a hyperbolic dependence of the same kind as expressed in Michaelis-Menten kinetics or in the Monod-equation of bacterial growth. The mathematical counterparts of the Michaelis-Menten and the Monod constant are given explicitly. As the Monod model is well-known in the Microbiology community, we think the transfer of these hyperbolic laws to the calorimetric description of the nutrient dependence of metabolic activity has become much more understandable.

4.)        Please improve the quality of figures 2-3. The dotted sub-functions are hard to see.

resp.:   The dotted lines have been replaced by solid lines to improve visibility.

- 153Eu decays into Gd and Sm. Wouldn't the toxicity of the decay products have to be taken into account, or is the concentration or the decay rate simply too small? I think this point deserve a clarifying sentence.

resp.:   The decay time of Eu152 is slightly longer than 13.5 years, such that the accumulation of decay products during the measureing time of days is indeed vanishingly small (less than nM concentration after 14 years). This is stated on lines 434-436.Minor points:

Maybe just a matter of taste: The terms "thermogram, thermal power" are unusual. The typical terms in literature are heat flow, heat flux or heat production rate.

resp.:   We understand the referee’s reasoning. On the other hand, for consistency of nomenclature with ref. 13, which describes the evaluation method, we have decided to keep the wording unchanged in the present manuscript.

9: Delete „-„

            done

L 39: Don't the decay products also have an effect on the overall toxicity?

resp.:   see response to point 4)

L 46: A comma is missing between alpha particles and protons.

resp.:   sentence has been amended

L67: Delete ii

            done

L70: Delete iii)

            done

L95: H deep 2 O

            done

L98: “4 mL”. You should explain why you have chosen this amount. It is probably to exclude an air space with the respective oxygen diffusion. To ensure anaerobic metabolism.

resp.:   The reviewer is right that cell division is not supported under these conditions. In fact, the experiment aimed at observing metabolic decline upon nutrient consumption in the absence of growth which results in a simpler nutrient / activity relation than for bacteria where the cell number increases over time. The experimental conditions have been adapted from previously published work on B. napus calorimetry which showed that, nevertheless, metal toxicity can be studied under these conditions (ref. 18). A large ampoule volume was chosen to enhance the signal at the expense of shortening the aerobic phase of metabolism.

            In the context of the present work, we like to demonstrate that the concept of isotope editing is in principle transferrable to eukaryotes and to other radionuclides. It is beyond the scope of the manuscript to draw more specific conclusions from the two additional systems (B. napus and Lysinibacillus sphaericus) which, therefore, are only mentioned in the Supplemental Information.

L 122: A space is missing.

            corrected

L124: “4 mL” for the experiments with plant cells? Typically, plant cells needs oxygen for growth.

            see above

Section 2.2: It is not clear to me what the experiments with the plant cells are needed for, which are barely discussed in the main article. I would omit the section.

res.:     As we like to present proof of principle for isotope-editing in a eukaryotic system, we prefer leaving the experimental data in the Supplemental Information and keeping a very short note on these data at the end of the discussion.

L186: Eq. (2): The variable “h” is not explained.

resp.:   The variable has now been explained (line 216)

 L210: Legend to Figure 2: Please provide the error range of the different initial growth rates.

resp.:   The error margins and their calculation have been included in the figure legend.

L223: Omit “)”

            done

L270, Fig. 4: Please provide the unit for the y-axis.

            has been included

L 367: “6 nJ s-1 “ please add the data in W (6 nW) because the calorimeter measure in W.

resp.:   The unit nJ has been omitted.For completeness, the distinction between radiative and non-radiative decay routes of the radionuclide is given in slightly more detail for the dose rate approximation (lines 423-427)

Supplementary Material: Unfortunately, I got an error message (Error 404 – File not found) and could therefore not review the SI.

Submission Date

31 December 2022

Date of this review

17 Jan 2023 13:59:12

Reviewer 2 Report

In the paper entitled Distinct Effects of Chemical Toxicity and Radioactivity on Metabolic Heat of Cultured Cells Revealed by 'Isotope-Editing', the authors describe a new methodology to study the effects of radiation on cell metabolism. Although the work is interesting, some things should be clarified

Major

1) I think it would be ideal to have a paragraph in which the microorganisms used are listed, indicating their temperatures, culture media, characteristics and why they were chosen

2) since the mic for Eu varies from microorganism to microorganism, are the authors sure that the values used in the experiments are below the MIC?

3) similar discourse with regard to uranium

4) If in the caption of figure 1, reference is made to figure 2, perhaps it would be ideal to present figure 2 first and then figure 1

5) since the differences, as the authors themselves say, are minimal, it would be better if they were clearly marked on the figure, because I do not see any significant effect at the moment

6) also in Figure 2-3 it would be ideal to highlight the differences, although in this case they are more pronounced

Minor:

line 19 remove the - in monitoring

line 127 between 50 (??) and 35 uM, to be rectified

Author Response

RESPONSE TO REVIEWER 2

We thank the reviewer for advising on experimental and conceptual aspects of our work. We have made revisions accordingly and answer to the comments point by point (resp.:) in the original text provided by the referee as detailed below.

REVIEWER 2

Open Review

English language and style

( ) English very difficult to understand/incomprehensible
( ) Extensive editing of English language and style required
( ) Moderate English changes required
( ) English language and style are fine/minor spell check required
(x) I don't feel qualified to judge about the English language and style

Yes

Can be improved

Must be improved

Not applicable

Does the introduction provide sufficient background and include all relevant references?

(x)

( )

( )

( )

Are all the cited references relevant to the research?

(x)

( )

( )

( )

Is the research design appropriate?

( )

(x)

( )

( )

Are the methods adequately described?

( )

(x)

( )

( )

Are the results clearly presented?

( )

(x)

( )

( )

Are the conclusions supported by the results?

(x)

( )

( )

( )

Comments and Suggestions for Authors

In the paper entitled Distinct Effects of Chemical Toxicity and Radioactivity on Metabolic Heat of Cultured Cells Revealed by 'Isotope-Editing', the authors describe a new methodology to study the effects of radiation on cell metabolism. Although the work is interesting, some things should be clarified

Major

1) I think it would be ideal to have a paragraph in which the microorganisms used are listed, indicating their temperatures, culture media, characteristics and why they were chosen

resp.:   we have included paragraphs with culture conditions separately from the calorimetric methods.

2) since the mic for Eu varies from microorganism to microorganism, are the authors sure that the values used in the experiments are below the MIC?

resp.:   Yes, Figure 1 is shows that 100 µM Eu barely affected growth of the culture and is thus far below the MIC of Eu for the Lactococcus strain.

3) similar discourse with regard to uranium

resp.:   A minimum inhibitory concentration of uranium for Brassica napus cells was not measured. However, the total uranium concentration of 50 µM U is lower than the half maximal inhibition concentration of U(VI) for B. napus cells, which was estimated by correlation of the oxidoreductase activity of the cells with IMC measurements (ref. 18). This stated in lines 117-119.

4) If in the caption of figure 1, reference is made to figure 2, perhaps it would be ideal to present figure 2 first and then figure 1

resp.:   We have omitted the reference to Figure 2, instead the legend in Figure 2 references Figure 1 to express the relation between the two ways of presenting the same data. Since Figure 1 displays the raw data and serves as an initial control to ensure marginal growth effects of the 100 µM background concentration (i.e., far below the MIC), we think that the sequence of the Figures follows the conceptual experiment design better than the reverse order.

5) since the differences, as the authors themselves say, are minimal, it would be better if they were clearly marked on the figure, because I do not see any significant effect at the moment

resp.:   The purpose of the Figure is to show the virtual absence of growth effects of the background concentration. The little influence is only visible upon full curve fitting and at higher Eu concentrations as shown in Figure 4. However, small features are now indicated by arrows and the slight decrease in peak metabolic activity in the presence of Eu is emphasized by corresponding number labels in Figure 1 and 2 and is further stated in the text (line 174-178) and the legend of Figure 2.

6) also in Figure 2-3 it would be ideal to highlight the differences, although in this case they are more pronounced

resp.:   It is hard for the eye to discern curvature changes which are revealed only by a full curve fitting procedure which determines the growth parameters. However, the successive decrease in peak metabolic activity has been labeled in the revised Figure 2 as it is directly visible.

Minor:

line 19 remove the - in monitoring

            done   

line 127 between 50 (??) and 35 uM, to be rectified

            done (line 131)

Submission Date

31 December 2022

Date of this review

18 Jan 2023 07:27:56

Author Response

We thank the reviewer for the full appreciation of our work and acceptance of the manuscript in its original version. Therefore, no further action was undertaken in response to this referee.

Round 2

Reviewer 1 Report

I am fine with your changes and have the feeling that the manuscript is now mature for publication.

Reviewer 2 Report

I thank the authors for responding to my comments, nevertheless, I believe an investigation into MIC should be done, this may not be so relevant to this manuscript, but perhaps it can be a starting point for future work.

Finally, I suggest increasing the resolution of all figures